# EVALUATING BLACK-BOX VULNERABILITIES WITH WASSERSTEIN-CONSTRAINED DATA PERTURBATIONS

**Adriana Laurindo Monteiro** *
Independent Researcher
mlaurindodrica@gmail.com

**Jean-Michel Loubes**
INRIA Regalia & ANITI
Institut de Mathématiques de Toulouse, France
Centre INRIA de l'Université de Bordeaux
jean-michel.a.loubes@inria.fr

## ABSTRACT

The growing use of Machine Learning (ML) tools comes with critical challenges, such as limited model explainability. We propose a global explainability framework that leverages Optimal Transport and Distributionally Robust Optimization to analyze how ML algorithms respond to constrained data perturbations. Our approach enforces constraints on feature-level statistics (e.g., brightness, age distribution), generating realistic perturbations that preserve semantic structure. We provide a model-agnostic diagnostic bench that applies to both tabular and image domains with solid theoretical guarantees. We validate the approach on real-world datasets providing interpretable robustness diagnostics that complement standard evaluation and fairness auditing tools.

## 1 INTRODUCTION

Modern machine learning models often function as black boxes whose reliability under data distribution shift remains difficult to assess. In safety-critical applications, it is essential to stress such models by exposing them to plausible variations in data distribution and observing the impact on predictions.

We propose a testing bench for black-box models on tabular and image data that systematically generates constrained distributional perturbations to evaluate model robustness. Our approach draws on concepts from optimal transport and distributionally robust optimization (DRO). In particular, we use the Wasserstein distance – a metric on probability distributions that measures the "cost" of transforming one distribution into another – to bound the magnitude of distribution shifts and ensure they remain realistic. By restricting attention to perturbations within a Wasserstein ball around the original data-generating distribution, we generate plausible stress scenarios that respect the underlying data structure (e.g. preserving relationships between features) while probing the model's weaknesses. This yields a principled test bench for model robustness, complementary to pointwise adversarial attacks.

Our Wasserstein perturbation framework produces distributional shifts that are interpretable and controllable. For example, one can stress a credit risk model by increasing the marginal distribution of "debt-to-income ratio" within a certain range, or test a medical diagnosis model under an aging population distribution — all while constraining the shift's overall Wasserstein distance to maintain plausibility. These perturbations are obtained by solving optimization problems that find the worst-case or most influential distribution within an allowed Wasserstein radius (an approach inspired by distributionally robust optimization (Azizian et al., 2023), (Sinha et al., 2018b)).

Unlike classical global sensitivity analysis which varies inputs in a Monte Carlo fashion, our method directly perturbs the input data distribution itself. This provides a global view of model sensitivity: rather than asking how a single prediction changes when an input is tweaked, we ask how the model's overall performance or output distribution changes under a small but targeted shift in the data-generating distribution. By casting this as an optimal transport problem, we leverage efficient algorithms and theoretical guarantees from the optimal transport literature (Blanchet & Murthy,

---

*Corresponding author

2019). The use of Wasserstein distance further ensures that stress test scenarios are constrained and realistic: because Wasserstein distance accounts for the geometry of the data space, a small perturbation (within a given radius) corresponds to a distribution shift that is limited in magnitude for each feature, avoiding implausible changes to the data (unlike unconstrained adversarial perturbations that might be unnatural in tabular data).

**Contributions** Our contributions stand on a strong theoretical and empirical foundation. First, we formalize a model-agnostic testing bench for black-box predictors using realistic distribution shifts via constrained optimal transport. Then, we develop a nontrivial theoretical framework, giving not only the explicit formula for the projected distributions but also its asymptotic behavior and a dual formulation that connects the projection problem to DRO and that can be of independent interest. Finally, we validate our stress framework across tabular and vision domains[1], demonstrating consistent and interpretable perturbation experiments on real-world datasets; we also include a fairness-relevant sensitivity analysis in the classification setting.

## 2 RELATED WORK

**Distributionally robust optimization (DRO).** The investigation formalized by DRO proposes to minimize the worst-case risk over all distributions within a certain distance (ambiguity set) of the training distribution (Wozabal, 2012). In (Mohajerin Esfahani & Kuhn, 2018) they show that optimization over a Wasserstein ball yields tractable reformulations and strong out-of-sample performance guarantees. Similarly, (Blanchet & Murthy, 2019) develop an optimal-transport based measure of distributional model risk. Other divergences and uncertainty sets have also been considered (e.g. $\phi$-divergences (Namkoong & Duchi, 2017), moment-based ambiguity sets (Delage & Ye, 2010), etc.). Recent studies have underscored the importance of evaluating models under domain shifts as for instance (Calderon et al., 2024) or (Chen et al., 2021).

**Global Sensitivity Analysis (GSA) and Stress Testing.** GSA techniques (e.g. variance-based methods) assess how uncertainty in input distributions contributes to uncertainty in outputs. Methods aim to handle correlated inputs and higher-order effects. Classic references such as (Sobol', 2001) and (Saltelli et al., 2008) provide frameworks for quantifying the influence of input variability on model output. They have been used in explainability and fairness in (Bénesse et al., 2024), (Jourdan et al., 2023) or (Fel et al., 2021). Stress testing originated in finance and risk management, where regulators ask "what if" the economy undergoes specific shocks. The work (Breuer & Csisz'ar, 2013) introduces a systematic approach to scenario stress tests by selecting distributions that maximize portfolio loss subject to a plausibility constraint measured by relative entropy.

**Explainability via Distributional Perturbations.** Finally, our approach connects to emerging work in explainable AI that moves beyond local, pointwise explanations (like SHAP (Lundberg & Lee, 2017) or LIME (Ribeiro et al., 2016)) towards global explanation frameworks. The work (Bachoc et al., 2023) introduced an entropic variable projection framework to explain model behavior through its response to changes in the input data distribution, rather than individual inputs. Quantile-constrained Wasserstein projections (Il Idrissi et al., 2024) studies model stability, generating distributions where certain feature quantiles are shifted and evaluating the resulting change in predictions.

## 3 METHODOLOGY

Our explainability framework is fundamentally grounded on a counterfactual point of view: we want to understand how model predictions would change if the inputs were modified. Perturbing a dataset we generate alternative scenarios that may cause different outputs, which enables us to answer what-if questions. When these perturbations are performed in a constrained manner, we can isolate the impact of a certain feature and thus obtain a global interpretation of the algorithm.

This methodology is translated into a mathematical formalism by a projection problem with respect to the Wasserstein distance. Let $\Omega \subset \mathbb{R}^d$ be a compact set. That is the set in which the data points reside. Denote by $\mathcal{P}(\Omega)$ and $\mathcal{M}(\Omega)$ respectively the space of probability and finite measures on $\Omega$. Consider a distance $d$ in $\Omega$, $p \in [1, +\infty[$ and $P, Q \in \mathcal{P}(\Omega)$. Define the $p$-Wasserstein distance

---

[1]Code available at `https://github.com/drica-monteiro/wasserstein_projection`

between $P, Q$ as

$$\mathcal{W}_p^p(P, Q) = \min_{\pi \in \Pi(P,Q)} \int d(x,y)^p d\pi(x,y). \tag{1}$$

The coupling set $\Pi(P, Q)$ denotes the set of distributions with marginals $P$ and $Q$. We are interested in the case $p = 2$ and $d(x, y) = \|x - y\|$ the Euclidean distance. If $P$ and $Q$ represent two distinct populations, the Wasserstein distance measures the minimal cost in transforming one population into the other.

In order to define the target set onto which the data points are projected, for a given $k \geq 1$, let $\Phi \colon \Omega \to \mathbb{R}^k$ be a continuous function representing a constraint and define $f \colon \mathcal{M}(\Omega) \to \mathbb{R}$, $f(P) = \mathcal{W}_2^2(P, Q)$ and $g(P) = \int_\Omega \Phi(x) dP(x) - t$, for fixed $t \in \mathbb{R}^k$ and $Q \in \mathcal{P}(\Omega)$. The set $D = \{P \in \mathcal{M}(\Omega) \mid \int_\Omega \Phi(x) dP(x) = t\} = g^{-1}(\{0\})$ is closed, for the weak convergence topology, and convex since $g$ is linear in $P$. The function $f$ is convex as it is the supremum of linear functionals by Kantorovich duality (Santambrogio, 2015), therefore the following projection problem is well defined

$$\min f(P) \text{ such that } P \in D. \tag{P}$$

The described approach has two essential aspects: flexibility and realism. First, we can control the value of a feature in a population whose distribution $P$ belongs to the set $D$ by calibrating the parameter $t$. Second, by projecting $Q$ onto this set $D$ we ensure that the modified population remains statistically plausible. As a result, the generated counterfactual explanations are both diverse and faithful to the original data.

**Notation.** We use a parameter $k \geq 1$ to denote the dimension of the calibration parameter $t \in \mathbb{R}^k$ and the feature dimension is $d$. The number $n$ denotes sample size. The Lagrange multiplier for problem equation P is represented by $\lambda \in \mathbb{R}^k$. The push-forward measure $T_\# Q$ is defined through $T_\# Q(A) := Q(T^{-1}(A))$ for every measurable set $A$. The inner product is denoted by $u^\top v = \sum_{i=1}^d u_i v_i$ for $u, v \in \mathbb{R}^d$. For a function $f : \mathbb{R}^d \to \mathbb{R}$ the gradient vector and the hessian matrix are denoted respectively by $\nabla f(x)$ and $D^2 f(x)$. $e_i$ is the $i$−th vector of $\mathbb{R}^d$.

## 4 THEORETICAL ANALYSIS

Our theoretical investigation is based on duality theory in normed spaces (Peypouquet, 2015) and the differential properties of the Wasserstein distance (Santambrogio, 2015). Proofs and additional results can be found in Appendix C and preliminary results on convex analysis can be checked in D.

### 4.1 PROJECTION THEOREM

**Theorem 1.** *For $y \in \mathbb{R}^d$, let $T_\lambda(y) \in \arg\min_x \{\|x - y\|^2 - \lambda^\top \Phi(x)\}$. Then $P^*$ is an optimal solution to problem equation P if, and only if, it is defined as $P^* = T_{\lambda^*} \# Q$ where $\lambda^* \in \mathbb{R}^k$ satisfies*

$$t = \int \Phi(T_{\lambda^*}(y)) dQ(y).$$

The constraint on problem equation P can be modified to include an inequality condition. This is detailed in Proposition 11 deferred to appendix.

**Remark 2.** *To guarantee existence and uniqueness of the optimal solution $P^*$, fix $y \in \mathbb{R}^d$ and define $h(x) = \|x - y\|^2 - \lambda^\top \Phi(x)$. For the function $T_\lambda$ be well defined, we need the set $\arg\min_{x \in \mathbb{R}^d} h(x)$ to be non-empty. Since the squared euclidean norm is continuous and coercive, a sufficient condition for existence of a minimum is coercivity of $-\Phi_i$ for $i = 1, \ldots, k$ (they already are continuous). A sufficient condition for uniqueness of the optimal solution of problem equation P is to impose strong convexity of the function $h$. Hence, the minimum of $h$, if it exists, is unique.*

### 4.2 CONSISTENCY

A key property of our approach is the continuity of the Wasserstein projection onto $D$ under weak convergence. Defining the following projections we can prove consistency:

$$P_n^* \in \arg\min_{P \in D} \mathcal{W}_2^2(P, Q_n) \text{ and } P^* \in \arg\min_{P \in D} \mathcal{W}_2^2(P, Q). \tag{2}$$

**Theorem 3.** *Consider a sequence $Q_n \xrightarrow{w} Q$ and the probabilities $P_n^*$ and $P^*$ defined above in equations 2. Then, $P_n^* \xrightarrow{w} P^*$.*

The applications of our work are based on the choice of $Q = \frac{1}{n} \sum_{i=1}^n \delta_{Z_i}$ as the empirical measure of a dataset $\{Z_i\}_{i=1}^n$. Therefore, the previous result states that the performance of the framework improves with more datapoints.

**Example 4.** *If $(Z_i)_{i \in \mathbb{N}}$ is an i.i.d sequence distributed according to $P$, the empirical measure $Q_n = \frac{1}{n} \sum_{i=1}^n \delta_{Z_i}$ converges weakly to the true measure $P$ by the strong Law of Large Numbers. Therefore, $P_n^* = T_{\lambda \#} Q_n \xrightarrow{w} T_{\lambda \#} Q = P^*$.*

### 4.3 PRACTICAL COMPUTATIONS

Depending on the constraint function $\Phi$ one is able to solve explicitly

$$T_\lambda(y) := \arg \min_{x \in \mathbb{R}^d} \left\{ \|x - y\|^2 - \lambda^\top \Phi(x) \right\} = \arg \min_{x \in \mathbb{R}^d} \mathcal{H}(x, y) \tag{3}$$

and give the projected distribution. Here we discuss the linear case that is used in the experiments and other examples are referred to Appendix E. When the computation of equation 3 is impractical to solve explicitly, we propose a method explained in Appendix A.

Suppose $k \leq d$ and $\Phi(x) = (x_1, x_2, \ldots, x_k)$. In this case, $\Phi$ is strongly convex. If $\lambda_i < 0$ for all $i = 1, \ldots, k$ then $h(x) = \|x - y\|^2 - \sum_{i=1}^k \lambda_i \Phi_i(x)$ is strongly convex and we have existence and uniqueness of

$$\inf_x \left\{ \|x - y\|^2 - \sum_{i=1}^k \lambda_i \Phi_i(x) \right\}$$

immediately guaranteed by finding a critical point. Let $h(x) = \|x - y\|^2 - \sum_{i=1}^k \lambda_i x_i$ and take derivative in $x$, $\nabla h(x) = 2(x - y) - \sum_{i=1}^k \lambda_i e_i$. The optimal $T_\lambda(y)$ is $T_\lambda(y) = \frac{\sum_{i=1}^k \lambda_i e_i}{2} + y$.

**Corollary 5.** *(Linear case) Suppose $Q = \frac{1}{n} \sum_{i=1}^n \delta_{Z_i}$ and $\Phi(x) = x_1$. Then $P^*$ is an optimal solution to problem equation P if, and only if, it is defined as $P^* = \frac{1}{n} \sum_{i=1}^n \delta_{Z_i + \lambda^* e_1/2}$ where $t = \frac{1}{n} \sum_{i=1}^n (Z_i^1 + \lambda^*/2)$.*

Concerning image data, with $d = p \times p$ pixels, choose $\Phi(x) = \frac{1}{d} \sum_{i=1}^d x_i$. If we flatten each image to obtain a $d$-dimensional vector, the function $\Phi$ represents the average brightness of that image. With the same reasoning as in the linear case, we can project a set of images with an average brightness constraint.

**Corollary 6.** *(Brightness constraint) Suppose $Q = \frac{1}{n} \sum_{i=1}^n \delta_{Z_i}$ and $\Phi(x) = \frac{1}{d} \sum_{i=1}^d x_i$. Then $P^*$ is an optimal solution to problem equation P if, and only if, it is defined as $P^* = \frac{1}{n} \sum_{i=1}^n \delta_{Z_i + \frac{\lambda}{2d}}$ where $t = \frac{\lambda}{2d} + \frac{1}{n} \frac{1}{d} \sum_{i=1}^n \sum_{j=1}^d Z_i^j$.*

**Remark 7.** *Although we did not performed the experiments for contrast perturbation, we can easily extend our findings by choosing $\Phi(x) = \frac{1}{d} \sum_{i=1}^n (x_i - mean(x_i))^2$ or similar. In that case, we could investigate how pixel variance may impact the task of image classification.*

### 4.4 DUAL FORMULATION

Based on standard duality theory for convex programs, we proved a dual formulation for our projection problem equation P. As already mentioned, one can find conditions for the dual formulation of a general convex problem in (Peypouquet, 2015). Additional details and proofs can be checked in the Appendix D.3 and D.2.

**Theorem 8.** *The dual problem of equation P is*

$$\max_{\lambda \in \mathbb{R}^k} \inf_{\mathcal{P}(\Omega)} \left\{ \mathcal{W}_2^2(P, Q) - \lambda^\top \left( \int \Phi dP - t \right) \right\}$$

$$= \max_{\lambda \in \mathbb{R}^k} \left\{ \lambda^\top t + \int_\Omega \inf_{x \in \Omega} \{ \|x - y\|^2 - \lambda^\top \Phi(x) \} dQ(y) \right\}.$$

The dual formulation[2], Theorem 8, plays a key role in enabling efficient computation of the projection. While the closed-form solution can be directly applied in the cases explored here, the dual enables scalable optimization when $\Phi$ has a more complex structure or acts in high-dimensional spaces. In particular, the dual solves an optimization in $\lambda \in \mathbb{R}^k$, where $k \ll d$, significantly reducing complexity for high-dimensional feature spaces. This paves the way for future extensions to unstructured data such as text or high-resolution images.

## 5 EXPLAINING MODELS VIA WASSERSTEIN PROJECTION

In this section, we connect the theoretical framework to the experimental procedure. We apply Theorem 1 in the empirical setting where $Q = \frac{1}{n} \sum_{i=1}^{n} \delta_{Z_i}$ represents the dataset. In this case, the optimal solution $P^*$ corresponds to a push-forward of the empirical measure, meaning that each data point $Z_i$ is mapped to a perturbed version $T_{\lambda^*}(Z_i)$.

The calibration parameter $t$ in Theorem 1 specifies the desired value of the feature statistic (e.g., the mean of a given variable). In practice, we parametrize $t$, which allows us to generate a family of perturbed datasets by continuously varying the constraint.

Therefore, the experiments consist of computing the transformed dataset $\{T_{\lambda^*}(Z_i)\}_{i=1}^{n}$ for different values of calibration parameter $t$, and evaluating how the model predictions change under these controlled distributional shifts.

### 5.1 EXPLAINABILITY MEASURES

Our analysis can be divided in three steps. Let $Z_1, \ldots, Z_n$ denote the original dataset and $Z_1^*, \ldots, Z_n^*$ the projected one. As given by Theorem 1, $T_{\lambda^*}(Z_i) = Z_i^*$. The first step of our investigation is to train the models in $Z_1, \ldots, Z_n$, then compute $Z_1^*, \ldots, Z_n^*$, and finally compute the explainability measures for the projected dataset.

In the binary classification setting, we are going to compute the portion of predicted 1's, PP1. Concerning the multi-class classification task explored in the image dataset, with $q$ different categories, we consider the average positive classification within each class $j$, that is PP1$_j$, for $j = 1, \ldots, q$:

$$\text{PP1} = \frac{1}{n} \sum_{i=1}^{n} f(Z_i^*) \text{ and } PP1_j = \frac{1}{n} \sum_{i=1}^{n} \mathbb{1}_{\{f(Z_i^*)=j\}}. \tag{4}$$

Analyzing the evolution of these measures with respect to calibration parameter provides insight into the learned outputs. We can understand which variable produces the increase or decrease of labels equal to 1 in the case of classification. Plotting the explainability measure in equation 4 highlights the influence that a covariate had for the decision rule.

In particular, the monotonicity of the response with respect to the stress parameter $\tau$ provides a meaningful and interpretable signal of how model predictions evolve under controlled shifts. In this context, we interpret a feature as having a positive influence when its associated response curve increases with $\tau$. Furthermore, the relative position of the curves provides a notion of importance, as features whose curves consistently lie above others induce larger changes in the model output and can therefore be interpreted as more influential.

### 5.2 PARAMETRIZING PERTURBATIONS WITH QUANTILES

To capture the mixed behavior of different variables with different ranges we need to introduce a parametrization parameter for the calibration $t$. The perturbation procedures adopted in the experiments are detailed below.

If $X \in \mathbb{R}^d$ take $j_0 \in \{1, \ldots, d\}$ and define $\Phi(X) = X^{j_0}$. The idea is to compute the stressed mean as $t = m_{j_0} + \epsilon_{j_0,\tau}$, where $m_{j_0} = \frac{1}{n} \sum_{i=1}^{n} X_i^{j_0}$ and $\epsilon_{j_0,\tau}$ is parametrized (Bachoc et al., 2023) as

---

[2]This result is closely related to DRO based on Wasserstein distance. It has deep connections to adversarial training in Machine Learning (Sinha et al., 2018a).

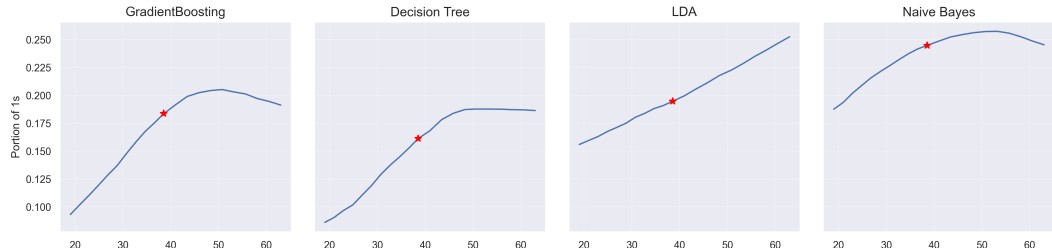

Figure 1: Impact of a feature in positive classifications: portion of predicted 1's as a function of mean *Age*. Red marker represents original dataset.

follows

$$\epsilon_{j_0,\tau} = \begin{cases} \tau(m_{j_0} - q_{j_0}(\alpha)), \text{ if } \tau \in [-1,0), \\ \tau(q_{j_0}(1-\alpha) - m_{j_0}), \text{ if } \tau \in (0,1], \\ 0, \text{ if } \tau = 0, \end{cases} \quad (5)$$

where $q_{j_0}(\alpha)$ is the $\alpha$- empirical quantile of the variable $X_{j_0}$ and $\alpha \in (0, 1/2)$. In that case, we have $q_{j_0}(\alpha) \leq m_{j_0} \leq q_{j_0}(1-\alpha)$ so the stressed mean satisfies $t \in [q_{j_0}(\alpha), q_{j_0}(1-\alpha)]$.

This parametrization not only allows one to understand the mixed behavior of different variables with different ranges but also improves realism and comparability across datasets and attributes. Specifically, the parameter $\tau$ parametrizes the amount of stress whatever the distribution of the $\{X_{j_0}^1, \ldots, X_{j_0}^n\}$ is. The stressed perturbation goes from the small quantile $q_{j_0}(\alpha)$, when $\tau = -1$, then pass to $m_{j_0}$, when $\tau = 0$, and becomes the large quantile $q_{j_0}(1-\alpha)$, when $\tau = 1$.

**Remark 9.** *Although we can not compute accuracy or any error metric since we have no true labels, we are able to check prediction stability under shifts. Details are discussed in Appendix B.2.*

## 6 EXPERIMENTS

The goal of our experiments is to analyze how different learning algorithms respond to controlled distributional shifts induced by Wasserstein projections. We follow a common protocol across all settings. First, each model is trained on the same training dataset. Second, we generate projected versions of the test dataset, parametrized by $\tau$. Finally, we analyze the model responses by computing and visualizing the proposed explainability criteria on the perturbed datasets. Throughout all experiments[3], we fix $\alpha = 0.05$ and consider 21 uniformly spaced values of $\tau$ in $[-1, 1]$. Additional analysis can be found in Appendix A, including the regression setting, comparison to other methods and computational complexity.

### 6.1 TWO-CLASS CLASSIFICATION

We consider the *Adult Income* dataset with $n = 48842$ observations with $d = 37$ features, randomly split in 80% for training and 20% for test. It describes numerical and categorical characteristics such as *Age*, *Capital Gain* and *Loss*, *Occupation*, *marital status*, *gender* and so on. Following Corollary 5, we want to understand what is the influence of each feature when classifying high income (more than 50000$ per year).

We trained Gradient Boosting, Decision Tree, LDA and Naive Bayes. As shown in Figure 1, the feature *Age* has a beneficial impact in positive classifications ($\geq$ 50k income) up to an age close to 50. On the other hand, observing Figure 2a, the impact of *Education-Num* is clear. As higher *Education-Num* is, the chance of having an income greater than 50k is bigger. Interestingly, the model Naive Bayes points out *Capital Gain* as the ntial variable: having a big amount of money in bank account is an obvious evidence of high income.

**Remark 10.** *We can also produce simultaneous shifts in multiple features in the same fashion taking $k > 1$. An example stressing Education-Num and Hours per week is plotted in Figure 2b. To*

---

[3]The choice of 21 values for $\tau$ is a visualization resolution choice and $\alpha = 0.05$ is a typical value widely used for instance in confidence levels, trimming and robustness computations.

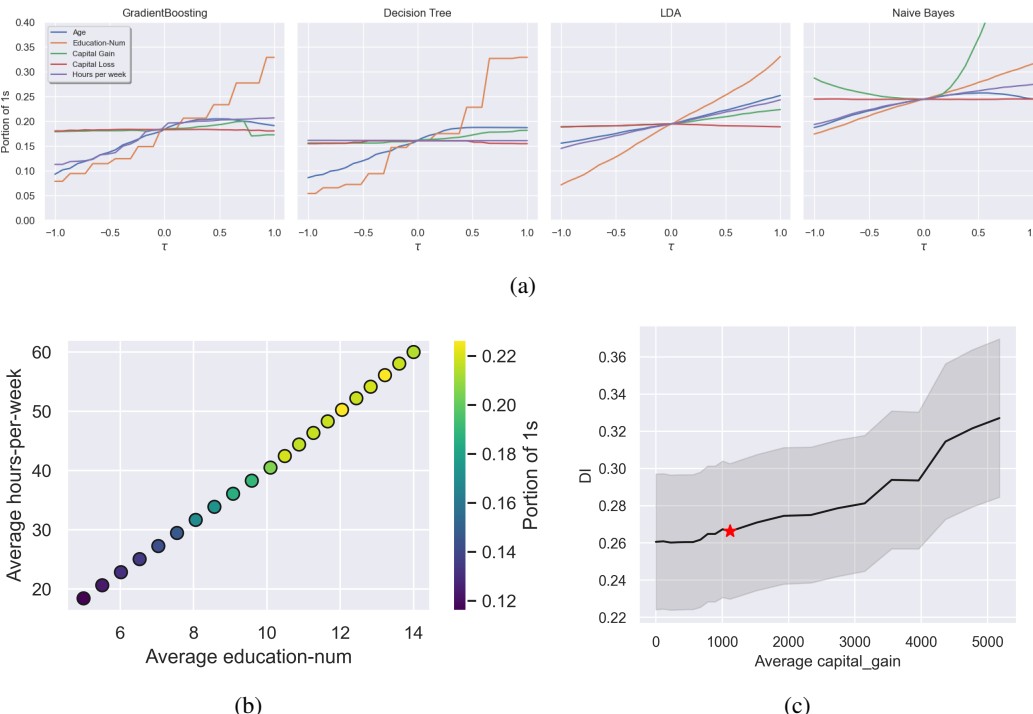

Figure 2: **(a)** Multiple mean changes impact on higher income ($\geq$50k). As $\tau$ increases (respectively decreases), the values of the selected feature increases (respectively decreases). $\tau = 0$ represents no projection. **(b)** Simultaneous mean perturbations for Gradient Boosting indicating a positive joint influence of features *Education-Num* and *Hours per week* on positive classifications. Color scale represents the proportion of predicted 1's. **(c)** Disparate Impact variation with respect to *Capital Gain* mean constraint. The red star represents the DI of the original dataset and the gray area denotes the confidence interval.

*handle more sophisticated constraints, it is a matter of solving the optimization problems discussed in Section 4.3. Detailed computations for different examples can be checked in Appendix E.*

### 6.1.1 FAIRWASHING

In the classification setting, an important measure to evaluate the fairness of a decision $Y \in \{0, 1\}$ made by an algorithm $g$ using the dataset $X$ is the Disparate Impact

$$\text{DI}(g) = \frac{\mathbb{P}\left(g(X) = 1 \mid S = 0\right)}{\mathbb{P}\left(g(X) = 1 \mid S = 1\right)}. \tag{6}$$

It quantifies how spread is the positive classification among the two subgroups given by the sensitive attribute $S$. It is desirable to have a $DI$ as close as possible to 1, meaning that the classifications are more or less similar between subgroups.

Here we highlight how the distribution shift may falsely indicate a fair algorithmic decision. After computing the projected datasets with constrained means as described in Section 5, compute the disparate impact equation 6 in each of these new datasets to investigate the impact of some features on this fairness criteria. We computed the Disparate Impact with a confidence interval as in (Besse et al., 2018). In our case, the sensitive attribute is gender, where $S = 0$ means female and $S = 1$ means male.

Figure 2c plots the impact of perturbing the feature *Capital Gain*. We can see that increasing the average *Capital Gain* also increases the fairness criteria. Note that the mean calibration is done regardless the feature *gender*. This process is the so-called *fairwashing*.

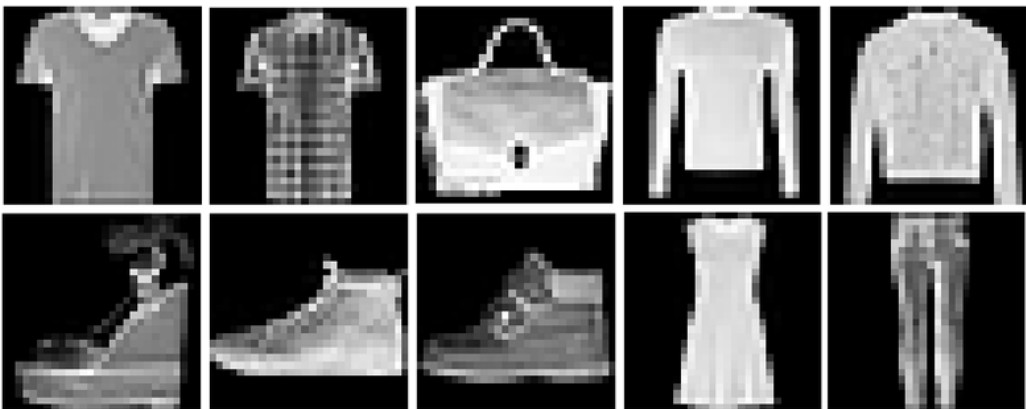

Figure 3: Random test sample of fashion classes. First row shows from left to right *T-shirt/top, Shirt, Bag, Pullover and Coat*. Second row displays *Sandal, Sneakers, Ankle Boots, Dress, Trouser.*

This phenomen is widely explored in (Lafargue et al., 2026). The distribution shift in a Wasserstein ball may create a representative-looking sample from an original distribution that falsely meet a fairness criterion, thereby masking the bias present in an algorithmic decision.

In this work, however, we are discussing a diagnostic tool. Our framework is designed to expose potential vulnerabilities, including scenarios where fairness metrics can be artificially improved. In this sense, our approach aims to reveal, rather than mitigate, such issues.

## 6.2 IMAGE CLASSIFICATION: BRIGHTNESS PERTURBATION ON FASHION-MNIST

Following Corollary 6, we investigate the impact of classifying an image after brightness stress. In our Wasserstein-constrained perturbation setting, this amounts to add a constant offset to all pixels, i.e. a global brightness shift. Note that we are not interested in 'pixel importance', but in showing how perturbing a fundamental aspect of an image, such as average brightness or contrast, can be relevant to a classifier output. Bear in mind that both brightness and contrast are global characteristics of an image set, which aligns well with our global explainability framework.

The dataset *Fashion-MNIST* (Xiao et al., 2017) consists of $n = 70000$ grayscale images with $d = 28 \times 28$ pixels representing 10 fashion categories such as *T-shirt/top, Trouser, Pullover, Dress, Coat, Sandal* and others.

We trained a small CNN with three convolutional blocks (390410 trainable parameters) and no normalization/standardization layer. We achieved $91\%$ test accuracy for 8 epochs, which provided a reasonably strong yet simple baseline for our stress-testing experiments.

We adopted the same procedure as before with one additional clipping step to ensure a valid pixel value. The explainability measure is the average positive classification within each class equation 4.

As shown in Figure 3, there are groups of classes where the intra-similarity is very high: *Sandals, Sneakers* and *Ankle Boots*,*T-shirt/top* and *Shirt*, *Pullover* and *Coat*. Finer details in the diagonal may differ a *Sandal* from a *Sneaker*, but if the *Sandal* has high hills, it looks like an *Ankle boot*. Therefore, perturbing the pixel intensity may confuse the model in classifying such classes. It also indicates how external factors such as lighting changes may affect a model output differently for each class. Inspecting Figure 4 we observe that phenomena: as $\tau$ gets close to 1, the model is less probable to classify a *Sandal* and a *Sneaker*.

On the other hand, the probability of classifying an *Ankle boot* increases. The opposite behavior also occurs for *Pullover* and *Coat*. As the *Coat* images get darker, the space between the arms becomes less distinguishable. It is interesting to see that increasing the brightness helps the model on classifying *Trouser*, which can be explained by the middle space between the legs. Brighter images drastically improve the classification of *Bags*, since the handle of the bag becomes clearer.

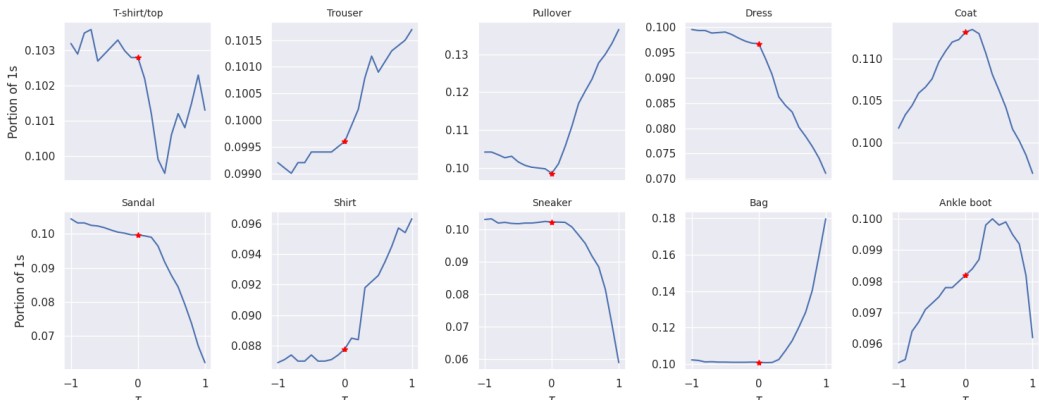

Figure 4: Portion of positive classifications for each class by perturbation parameter $\tau$. Red marker represents original dataset.

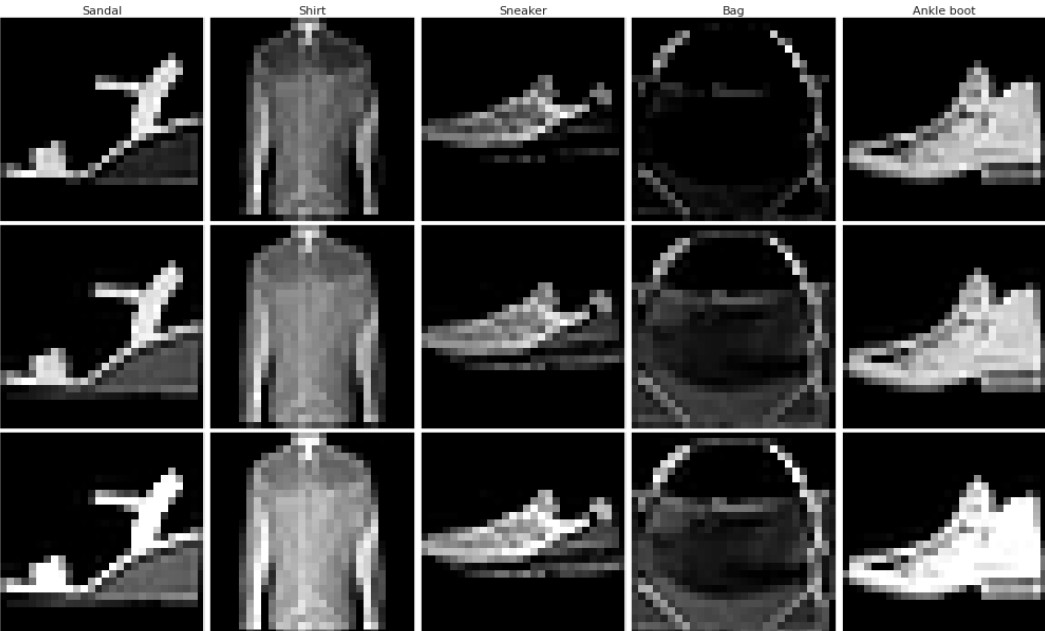

Figure 5: Perturbation of test sample of classes *Sandal, Shirt, Sneaker, Bag* and *Ankle boot* (from left to right). First row shows the darker perturbation ($\tau = -1$), middle row is the original image ($\tau = 0$) and third row is the brightest perturbation ($\tau = 1$).

In Figure 5 we plotted a random sample of test images to compare the original ($\tau = 0$) with the darker ($\tau = -1$) and brighter ($\tau = 1$) image.

## 7   CONCLUSION

**Discussion.** Our approach provides counterfactual explanations by generating synthetic data points that satisfy user-specified constraints while remaining close to the original data distribution, ensuring in-distribution plausibility. Unlike naive interventions that produce out-of-distribution artifacts or standard feature-importance methods that rely on local attributions or aggregate error measures, our framework enables controlled and feasible distributional modifications that isolate and quantify feature effects at the population level.

In contrast to resampling-based approaches, which only reweight observed data, our method generates genuinely new synthetic individuals, allowing a richer exploration of counterfactual scenarios. We further extend the framework to image data through pixel-level perturbations, though we note that optimal transport in high-dimensional pixel spaces is computationally expensive and may not fully capture the underlying geometric structure of images.

Finally, we propose a general methodology to craft stress-test distributions, with Wasserstein-distance constraints ensuring that each crafted distribution is a realistic variant of the original. By explicitly studying the impact of actual feature modifications under distributional plausibility constraints, our framework yields interpretable and actionable explanations. Besides that, our framework does not correct the vulnerabilities it reveals; it is a diagnostic tool meant to complement robust training.

**Limitations.** In the binary classification and regression setting we chose tree-based algorithms and generative models because they are insensitive to feature scaling. Since the Wasserstein projection corresponds to shifting the data, our analysis would be compromised in algorithms that require standardization before training. Despite that limitation, there is no concern in the setting where the function $\Phi$ does not results in shifting the dataset.

Our framework does not yet provide explicit quantitative criteria to systematically characterize feature influence behaviors. Future work could formalize these notions by introducing metrics that capture monotonicity, total variation, or curvature of the response curves, as well as quantitative measures of fairness distortion based on the variation of fairness indicators across $\tau$.

**Future work.** We plan to reproduce the brightness perturbation scheme in classification tasks that shows disparities across demographic attributes such as gender and ethnic origin. Additionally, we aim to apply our modification scheme in a learned embedding space, or in a factorized concept space (Fel et al., 2021). That approach is also suitable for unstructured data, such as text. Computing counterfactuals in such spaces would provide an alternative to commonly used importance indicators such as SHAP values or Sobol indices, and would yield genuinely interventional explanations at this level of concepts.

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

## A   EXTENDED PRACTICAL COMPUTATIONS

Here we propose a method to compute the solution of the optimization program 3 when there it has no closed form by hand, as explained in Section 4.3. We can replace the exact solution by an approximation obtained using a gradient descent. Additional details are provided in the appendix.

$$x^{t+1} = x^\top - \eta_t \nabla \mathcal{H}(x^t)$$
$$\nabla \mathcal{H}(x) = \nabla \|x - y\|^2 - \lambda^\top \nabla \Phi(x)$$

It thus requires to be able to compute the quantity $\nabla \Phi(x)$. Note that in the difficult case where the constraint is laid on the neural network $\Phi(x) = f(x)$ (maybe the mean of the scores), using that

$$\underbrace{\nabla_x \mathcal{L}(f(x), y) := (\nabla_y \mathcal{L}) \frac{\partial f}{\partial x}}_{\text{backpropagate gradients}}$$

the gradient in the gradient descent algorithm can be approximated, as soon as we have stored the gradients during the learning phase.

## B   ADDITIONAL EXPERIMENTAL ANALYSIS

### B.1   REGRESSION

For the regression case, we are interested in the average and variance criterion

$$M = \frac{1}{n} \sum_{i=1}^{n} f(X_i^*) \text{ and } Var = \frac{1}{n} \sum_{i=1}^{n} (f(X_i^*) - M)^2. \tag{7}$$

The evolution of the mean criteria indicates how a change in a specific feature affects the output. The variance criteria on the other hand, shows stability with respect to projection of the dataset.

The *Boston Housing* dataset contains $n = 506$ observations and $d = 12$ features. Each point represents houses in Boston and the target is the price of the house to be sold. The variables we are interested in are *lstat, rm, dis, crim* and *nox*. Respectively, they represent percentage of lower status of the population, average number of rooms per dwelling, weighted distances to five Boston employment centers, per capita crime rate by town and nitric oxides concentration (parts per 10 million).

We trained Gradient Boosting and Random Forest in randomly chosen 80% of the observations and tested in 20% remaining ones. Two important curves draw the attention when looking at Figure 6: the red and the blue ones, related respectively to *lstat* and *rm*. It's reasonable that more rooms in a house make an increase in price. The feature *lstat* has the opposite influence: lower values of it forces a decrease in price. The other features are less influential, since there is no big change in the average price with different values of $\tau$. It is clear that these curves are almost horizontal. When looking at price variance, another feature seems to create some instability: *dis*. Houses closer to Boston employment centers, $\tau < 0$, have heterogeneous prices while houses far to Boston employment centers, $\tau > 0$, have less variability in prices.

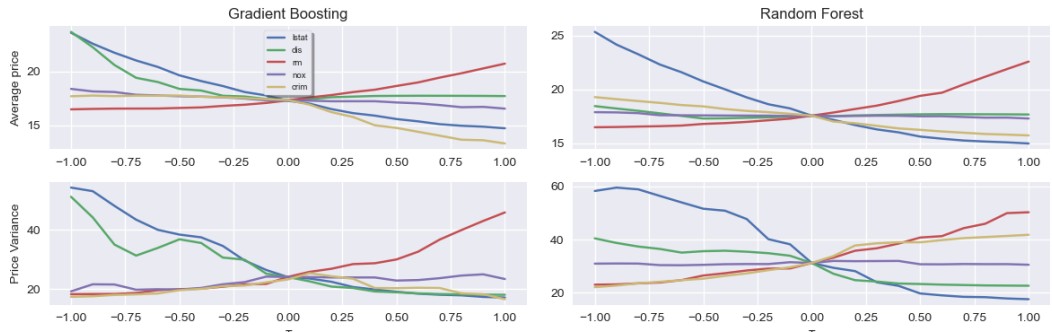

Figure 6: **(Top - Average price)** average of price predicted for projected datasets with respect to $\tau$. **(Bottom - Price variance)** with respect to $\tau$. There is no projection when $\tau = 0$. The higher (respectively lower) value for $\tau$, the higher (respectively lower) the value of the feature.

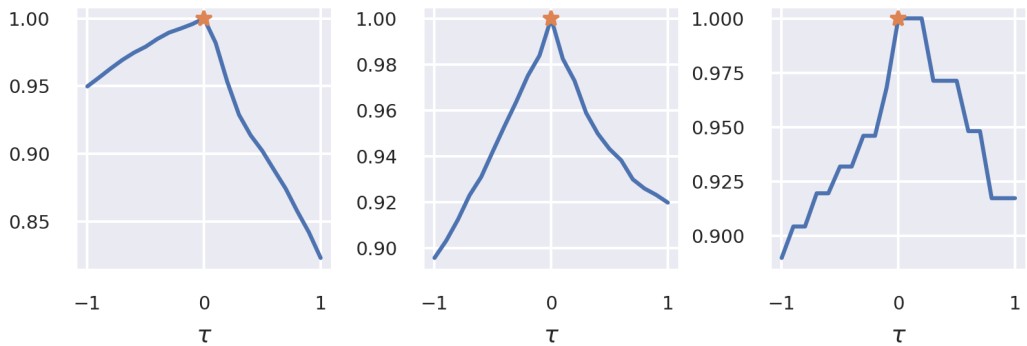

Figure 7: Percentage of unchanged predictions versus perturbation parameter $\tau$. Red marker represents original dataset.

## B.2 PREDICTION STABILITY UNDER SHIFT

Our framework also represents a robustness test with respect to mean constraint. Here we are comparing the labels of the projected data with the original one and plotting the percentage of unchanged predictions of the test set by perturbation parameter $\tau$. Inspecting the first plot of Figure 7, we can see that increasing the brightness level on the *Fashion-MNIST* experiment changes almost 20% of the labels while decreasing brightness does not represent great impact. The second and third curves, related to *Adult Income* dataset, represent respectively the prediction stability with respect to $\tau$ when perturbing the features *Age* and *Education-Num*. The experiments in Section 6.1 indicate both features as highly relevant to the model, which can be now confirmed as we can see that perturbing both of them (regardless the sign of $\tau$) results in changes of nearly 10% of the labels.

## B.3 EMPIRICAL COMPARISON TO BASELINE METHODS

For the purpose of getting a complete understanding of our framework, in this section we are going to consider other explainability tools: SHAP, entropic variable projection (Bachoc et al., 2023) and variable importance. Here we used (Lundberg et al., 2019) for the exact computation of SHAP values.

The first is not only the most popular method, but it can provide a local interpretation. The second also relies on projecting datasets therefore it is conceptually close to ours. Finally, variable importance in tree-based algorithms is a widely used approach for explainability.

We trained a XGBoost on the *Adult* dataset. The results are reported for the test set with $n = 9769$ samples.

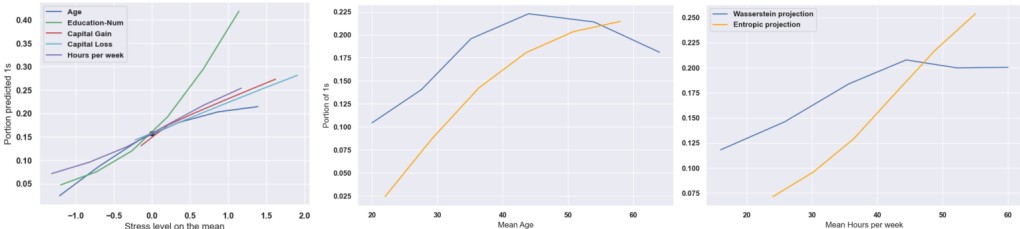

Figure 8: **(Left)** SHAP values of a specific individual **(Middle)** SHAP values averaged across the dataset in descent order. **(Right)** Feature importance score with importance weight.

Figure 9: **(Left)** Multiple mean stress obtained with entropic variable projection. **(Middle)** Stressing the feature *Age*. **(Right)** Stressing the feature *Hours per week*.

Figure 8 points to the difference between a global and a local approach to explainability. The figure on the left shows SHAP values computed for a single observation sorted in descent order. The feature *Capital Gain* contributed the most. This is in agreement with our framework for the Naive Bayes classifier, as showed by Figure 2a. On the other hand, when we use SHAP values to investigate the global behavior, the summary plot on the middle surprisingly shows *Marital Status* as the most relevant feature. On the right we observe a bar plot with variable importance computed using *weight* metric (the "weight" corresponds to the number of times a feature appears in a tree). Again, we obtain a different conclusion from our approach: the most important feature is *Age*. Although age can be a good proxy for having a high income, these experiments show how our approach produces a more reasonable explanation for an entire population.

Concerning the entropic projection, Figure 9 shows respectively from left to right the effect of multiple mean-constraint perturbations and the isolated effect of perturbing the features *Age* and *Education-Num*. From the first plot, we see that the entropic projection also shows the feature *Education-Num* as the ntial one and both frameworks show an increase in the proportion of predicted 1's as mean *Age* increases, but the Wasserstein projection shows a peak around middle age, which is a more realistic explanation. Analyzing the feature *Hours per week*, we see that both projections show a strong positive trend, while the entropic projections amplify the effect of mean shifts and Wasserstein projections act as a more conservative perturbation mechanism.

These results show that mean-constrained distributional perturbations can substantially alter classifier behavior, and that the geometry of the projection (Wasserstein vs. entropic) plays a critical role in shaping both sensitivity and fairness-related outcomes.

### B.4 COMPUTATIONAL COMPLEXITY OF THE METHOD

The proposed method is constituted by three main steps: training a model $f$ on a set $\{X_i\}_{i=1}^n$, projecting it to obtain $\{X_i^*\}_{i=1}^n$ and compute the explainability criteria. The first step inherits the computational complexity of the underlying algorithm. The projection step is composed by computing the Lagrange multiplier for the specific calibration parameter $t$ then solving equation 3. In the linear case, we need to compute the average per feature investigated and solve a linear system $k \times k$. Then, it is essentially linear in the sample size $n$, $\mathcal{O}(nk)$. Once the multiplier is known, we need to evaluate $T_\lambda$, that is, we need to update $k$ coordinates resulting in a complexity $\mathcal{O}(nk)$. Denoting

by $C$ the complexity of evaluating the model $f$, that has a complexity of $\mathcal{O}(nC)$. In the analysis of experiments in Section 6, we followed that approach for $N_\tau = 21$ different values of $\tau$. Therefore, the overall cost of our approach is $\mathcal{O}(nk)$, which is linear in the sample size $n$ and in the number of stressed features regardless of feature dimension $d$. In the general case, the complexity remains linear in $n$ but now the cost of evaluating the gradient of $\Phi$ might depend on feature dimension $d$.

## C  PROOFS AND ADDITIONAL RESULTS

### C.1  PROOF OF THEOREM 1

By Theorem 12, it is necessary and sufficient to find Lagrange multipliers $\lambda_1, \ldots, \lambda_k \in \mathbb{R}$ such that

$$\sum_{i=1}^{k} \lambda_i \nabla g_i(P^*) \in \partial f(P^*) \quad \text{(extremality condition)}$$

$$g(P^*) = 0 \quad \text{(feasibility)}$$

for a point $P^*$ be optimal for problem equation P. The subgradient of $f$ is given, Proposition 7.17 in (Santambrogio, 2015), by the set of Kantorovich potentials between $P^*$ and $Q$:

$$\partial f(P^*) = \left\{ \phi \in C(\Omega) \mid \int \phi dP^* + \int \phi^c dQ = \mathcal{W}_2^2(P^*, Q) \right\}.$$

Each functional $g_i$ is linear in $P$, then $\nabla g_i(P) = g_i$. Since the dual of $\mathcal{M}(\Omega)$ is $C(\Omega)$, we can say $\nabla g_i(P) = \Phi_i$. Now, notice that the definition of $T_\lambda$ implies

$$\mathcal{W}_2^2(Q, T_{\lambda\#}Q) \leq \int \inf_x \left\{ \|x - y\|^2 - \lambda^\top \Phi(x) \right\} dQ(y)$$

$$+ \int \lambda^\top \Phi(y) dT_{\lambda\#}Q(y)$$

$$= \int (\lambda^\top \Phi)^c(y) dQ(y) + \int \lambda^\top \Phi(y) dT_{\lambda\#}Q(y).$$

Strong duality of the Kantorovich problem, see (Santambrogio, 2015), guarantees that this inequality is indeed an equality.

Our calculations above prove the extremality condition for $P^* = T_{\lambda^\star\#}Q$. The feasibility condition imposes

$$t = \int \Phi(x) dP^*(x) = \int \Phi(T_{\lambda^\star}(y)) dQ(y). \tag{8}$$

### C.2  GENERALIZING THEOREM 1

**Proposition 11.** *Consider the following minimization problem*

$$\min \mathcal{W}_2^2(P, Q_n) \text{ s.t. } \int_\Omega \Phi(x) dP(x) \geq t. \tag{9}$$

*Then $Q_t$ is optimal for problem equation 9 if, and only if, it is defined as the push-forward*

$$Q_t = T_{\lambda^\star\#}Q_n$$

*where $T_\lambda(y) \in \arg\min_x \left\{ \|x - y\|^2 - \lambda^\top \Phi(x) \right\}$ and $\lambda^\star \in \mathbb{R}_{\geq 0}^k$ solves*

$$\int_\Omega \Phi(T_{\lambda^\star}(x)) dQ(x) \geq t$$

*and*

$$\lambda^{\star\top} \left( t - \int_\Omega \Phi(T_{\lambda^\star}(x)) dQ(x) \right) = 0.$$

## C.3 PROOF OF PROPOSITION 11

Let $g$ be the continuous function $g(P) = t - \int_\Omega \Phi(x)dP(x)$ and $f(P) = \mathcal{W}_2^2(P, Q)$. The set $\{P \in \mathcal{M}(\Omega) \mid \int_\Omega \Phi(x)dP(x) \geq t\} = g^{-1}([0, \infty))$ is closed for the weak convergence as $[0, \infty)$ is closed. Then the projection problem is well defined. Before applying the Lagrange multipliers theorem, we have to verify Slater's condition. By continuity of $\Phi_i$ and compacity of $\Omega$ we can consider, for $i = 1, \ldots, k$, $x_0^i \in \Omega$ such that $\Phi_i(x_0^i) = \min_{x \in \Omega} \Phi_i(x)$. Take $\alpha \in \mathbb{R}$ such that $\max_{1 \leq i \leq k} t_i/\Phi_i(x_0^i) < \alpha$. Then $\bar{P} = \alpha\delta_{x_0^i}$ satisfies $g_i(\bar{P}) < 0$ for $i = 1, \ldots, k$. The Lagrange multipliers theorem guarantees that $P^*$ is optimal for problem equation 9 if, and only if, there exists $\lambda_1, \ldots, \lambda_k \geq 0$ such that

$$\sum_{i=1}^k \lambda_i \nabla g_i(P^*) \in \partial f(P^*) \quad \text{(extremality condition)}$$

$$g(P^*) \leq 0 \quad \text{and } \lambda_i g_i(P^*) = 0, i = 1, 2, \ldots, k \text{ (feasibility)}$$

The proof of the extremality condition is completely analogous to the proof of Theorem 1. To conclude, we need to find $\lambda^\star \in \mathbb{R}_{\geq 0}^k$ such that the feasibility condition is satisfied:

$$t \leq \int_\Omega \Phi(T_{\lambda^\star}(x))dQ(x) \tag{10}$$

and

$$\lambda^{\star\top} \left( t - \int_\Omega \Phi(T_{\lambda^\star}(x))dQ(x) \right) = 0. \tag{11}$$

## C.4 PROOF OF THEOREM 3

The result follows by continuity of $P \mapsto \mathcal{W}_2^2(P, Q)$ with respect to weak convergence. Indeed, since the feasible set $D$ is closed for the weak convergence, the function $P \mapsto \inf\{\mathcal{W}_2^2(P, Q) \mid P \in D\}$ is also continuous. Therefore, if $Q_n \xrightarrow{w} Q$ we have

$$\inf_{P \in D} \mathcal{W}_2^2(P, Q_n) \to \inf_{P \in D} \mathcal{W}_2^2(P, Q).$$

Now suppose that $P_n^*$ does not converge to a minimizer of $\mathcal{W}_2^2(P, Q)$ in $D$. Since $\Omega$ is compact and $D$ is closed, we can admit a convergent subsequence $P_{n_k}^* \xrightarrow{w} P^*$. Note that

$$\mathcal{W}_2^2(Q, P^*) = \lim_k \mathcal{W}_2^2(Q_{n_k}, P_{n_k}^*) = \lim_k \inf_{P \in D} \mathcal{W}_2^2(P, Q_{n_k})$$

$$= \inf_{P \in D} \mathcal{W}_2^2(P, Q),$$

a contradiction.

## C.5 PROOF OF THEOREM 8

Theorem 15 guarantees

$$\inf_{P \in D} \mathcal{W}_2^2(P, Q) = \max_{\lambda \in \mathbb{R}^k, \psi_i^* \in D_i^*} \left\{ \sum_{i=1}^k \lambda_i g_i^*(\psi_i^*) - f^* \left( \sum_{i=1}^k \lambda_i \psi_i^* \right) \right\},$$

where the set $D_i^* = \{\psi^* \in C(\Omega) \mid g_i^*(\psi^*) < \infty\}$. It is a matter of rewriting equation equation 8 using Lemma equation 13 and equation equation 12:

$$\inf_{P \in D} \mathcal{W}_2^2(P, Q) = \max_{\lambda \in \mathbb{R}^k, \psi_i^* \in D_i^*} \sum_{i=1}^k \lambda_i t_i - f^* \left( \sum_{i=1}^k \lambda_i \psi_i^* \right)$$

$$= \max_{\lambda \in \mathbb{R}^k} \left\{ \lambda^\top t + \inf_{P \in \mathcal{P}(\Omega)} \left\{ -\lambda^\top \int_\Omega \Phi dP + W_2^2(P, Q) \right\} \right\}$$

$$= \max_{\lambda \in \mathbb{R}^k} \left\{ \inf_{P \in \mathcal{P}(\Omega)} \left\{ \lambda^\top (t - \int_\Omega \Phi dP) + W_2^2(P, Q) \right\} \right\}.$$

Completing the computation with Lemma equation 14:

$$\inf_{P \in D} W_2^2(P, Q) = \max_{\lambda \in \mathbb{R}^k, \psi_i^* \in D_i^*} \sum_{i=1}^k \lambda_i t_i - f^* \left( \sum_{i=1}^k \lambda_i \Phi_i \right)$$

$$= \max_{\lambda \in \mathbb{R}^k} \left\{ \lambda^\top t + \int_\Omega \left( \sum_{i=1}^k \lambda_i \Phi_i \right)^c dQ \right\}$$

$$= \max_{\lambda \in \mathbb{R}^k} \left\{ \lambda^\top t + \int_\Omega \inf_{x \in \Omega} \{ c(x, y) - \lambda^\top \Phi(x) \} dQ(y) \right\}.$$

# D  CONVEX OPTIMIZATION IN NORMED SPACES

## D.1  LAGRANGE MULTIPLIERS

Let $f, g_1, \ldots, g_m : X \to \mathbb{R} \cup \{+\infty\}$, denote proper, lower-semicontinuous, and convex functions along with continuous affine functions $h_1, \ldots, h_p : X \to \mathbb{R}$, where $X$ is a normed space.

We shall derive optimality conditions for the problem of minimizing $f$ over the set $C$ defined by

$$C = \{ x \in X : g_i(x) \le 0 \text{ for all } i, \text{ and } h_j(x) = 0 \text{ for all } j \},$$

assuming that this set is nonempty.

Since each $g_i$ is convex and each $h_j$ is affine, the set $C$ is convex. To simplify the notation, write

$$S = \operatorname{argmin}\{ f(x) : x \in C \}.$$

A qualification condition to derive our result is the following:

**Slater's Condition.** There exists $x_0 \in \operatorname{dom}(f)$ such that

$$g_i(x_0) <, i = 1, \ldots, m, \text{ and } \quad h_j(x_0) = 0, j = 1, \ldots, p.$$

**Theorem 12.** *If $\hat{x} \in S$ and Slater's condition holds, then there exist $\hat{\lambda}_1, \ldots, \hat{\lambda}_m \ge 0$, and $\hat{\mu}_1, \ldots, \hat{\mu}_p \in \mathbb{R}$, such that $\hat{\lambda}_i g_i(\hat{x}) = 0$ for all $i = 1, \ldots, m$ and*

$$0 \in \partial \left( f + \sum_{i=1}^m \hat{\lambda}_i g_i \right)(\hat{x}) + \sum_{j=1}^p \hat{\mu}_j \nabla h_j(\hat{x}). \tag{$\star$}$$

*Conversely, if $\hat{x} \in C$ and there exist $\hat{\lambda}_1, \ldots, \hat{\lambda}_m \ge 0$, and $\hat{\mu}_1, \ldots, \hat{\mu}_p \in \mathbb{R}$, such that $\hat{\lambda}_i g_i(\hat{x}) = 0$ for all $i = 1, \ldots, m$ and ($\star$) holds, then $\hat{x} \in S$.*

Note that our constraint in problem equation P is linear, so strong duality holds without the need to verify Slater's condition. Check chapter 3 in (Peypouquet, 2015) for more details.

## D.2  FENCHEL CONJUGATE OF WASSERSTEIN DISTANCE

Before stating the dual problem of equation P, we will see how to calculate the conjugate function of the objective and the constraint in problem equation P.

**Lemma 13.** *Consider $\Phi_i : \Omega \to \mathbb{R}$ and $t_i \in \mathbb{R}$. The Fenchel conjugate of*

$$g_i : \mathcal{M}(\Omega) \to \mathbb{R}, g_i(P) := \int_\Omega \Phi_i(x) dP(x) - t_i$$

*is*

$$g_i^* : C(\Omega) \to \mathbb{R}, g_i^*(\psi^*) := \begin{cases} t_i, & \text{if } \psi^* = \Phi_i \\ +\infty, & \text{otherwise.} \end{cases}$$

*Proof.* By definition of Fenchel conjugate,

$$g_i^*(\psi^*) = \sup_{P \in \mathcal{M}(\Omega)} \{\langle \psi^*, P \rangle - g(P)\}$$

$$= \sup_{P \in \mathcal{M}(\Omega)} \left\{ \int_\Omega \psi^*(x) dP(x) - \int_\Omega \Phi_i(x) dP(x) + t_i \right\}$$

$$= \sup_{P \in \mathcal{M}(\Omega)} \left\{ \int_\Omega \psi^*(x) - \Phi_i(x) dP(x) \right\} + t_i.$$

It is easy to see that the supremum is zero if $\psi^* = \Phi_i$. In the other case, we can choose $P_n = n\delta_{\bar{x}}$ where $\bar{x}$ is such that $\psi^*(\bar{x}) - \Phi_i(\bar{x}) > 0$ and make $n \to \infty$ to see that the supremum is $\infty$. $\qquad\square$

**Lemma 14.** *Fix $Q \in \mathcal{P}(\Omega)$. The Fenchel conjugate of*

$$f : \mathcal{M}(\Omega) \to \mathbb{R}, f(P) := \begin{cases} \mathcal{W}_2^2(P, Q), & \text{if } P \in \mathcal{P}(\Omega) \\ +\infty, & \text{otherwise.} \end{cases}$$

*is*

$$f^* : C(\Omega) \to \mathbb{R}, f^*(\psi^*) := -\int_\Omega (\psi^*)^c dQ.$$

*Proof.* By definition,

$$f^*(\psi^*) = \sup_{P \in \mathcal{M}(\Omega)} \langle \psi^*, P \rangle - f(P) = \sup_{P \in \mathcal{P}(\Omega)} \langle \psi^*, P \rangle - f(P)$$

$$= \sup_{P \in \mathcal{P}(\Omega)} \int_\Omega \psi^* dP - \mathcal{W}_2^2(P, Q) \tag{12}$$

$$= \sup_{P \in \mathcal{P}(\Omega)} \left\{ \int_\Omega \psi^* dP - \sup_{\varphi \in C(\Omega)} \int_\Omega \varphi dP + \int_\Omega \varphi^c dQ \right\}$$

$$= \sup_{\mathcal{P}(\Omega)} \left\{ \inf_{\varphi \in C(\Omega)} -\int_\Omega \varphi dP - \int_\Omega \varphi^c dQ + \int_\Omega \psi^* dP \right\}.$$

We can choose $\varphi = \psi^*$ and get the inner infimum equals to $-\int_\Omega \psi^{*c} dQ$:

$$f^*(\psi^*) = \sup_{\mathcal{P}(\Omega)} -\int_\Omega \psi^{*c} dQ$$

$$= -\int_\Omega \psi^{*c} dQ.$$

$\qquad\square$

### D.3 FENCHEL DUALITY THEOREM

Since the constraint in problem equation P is linear, we are going to use a less general dual formulation based on the following result of (Rockafellar, 1966):

**Theorem 15.** *If there exist Lagrange multipliers in the sense of Theorem 12, then*

$$\inf_{x \in C} f(x) = \max_{\lambda \in \mathbb{R}^k, \psi_i^* \in D_i^*} \left\{ \sum_{i=1}^k \lambda_i g_i^*(\psi_i^*) - f^* \left( \sum_{i=1}^k \lambda_i \psi_i^* \right) \right\},$$

*where*

$$D_i^* = \{\psi^* \in C(\Omega) \mid g_i^*(\psi^*) < \infty\}.$$

# E  EXAMPLES

## E.1  NORM

Suppose $\Phi(x) = \|x\|^2$. In this case, $k = 1$ and $\Phi$ is strongly convex. If $\lambda < 0$, then $h(x) = \|x - y\|^2 - \lambda\Phi(x)$ is strongly convex and we have existence and uniqueness of

$$\inf_x \left\{ \|x - y\|^2 - \sum_{i=1}^k \lambda_i \Phi_i(x) \right\}$$

immediately guaranteed by finding a critical point. Take derivative in $x$:

$$\nabla h(x) = 2(x - y) - 2\lambda x = 2x(1 - \lambda) - 2y. \tag{13}$$

The optimal $T_\lambda(y)$ is $T_\lambda(y) = y/(1 - \lambda)$.

**Corollary 16.** *(Norm) Suppose $\Omega$ is compact, $d^2(x, y) = \|x - y\|^2$, $Q = \frac{1}{n}\sum_{i=1}^n \delta_{Z_i}$ and $\Phi(x) = \|x\|^2$. Then $P^*$ is an optimal solution to problem equation P if, and only if, it is defined as $P^* = \frac{1}{n}\sum_{i=1}^n \delta_{Z_i/(1-\lambda)}$ where $t = \frac{1}{n}\sum_{i=1}^n \Phi(Z_i/(1 - \lambda))$.*

## E.2  QUADRATIC

Suppose $k \le d$ and $\Phi(x) = (x_1^2, x_2^2, \ldots, x_k^2)$. In this case, $\Phi$ is strongly convex. If $\lambda_i < 0$ for all $i = 1, \ldots, k$ then $h(x) = \|x - y\|^2 - \sum_{i=1}^k \lambda_i \Phi_i(x)$ is strongly convex and we have existence and uniqueness of

$$\inf_x \left\{ \|x - y\|^2 - \sum_{i=1}^k \lambda_i \Phi_i(x) \right\}$$

immediately guaranteed by finding a critical point. Let $h(x) = \|x - y\|^2 - \sum_{i=1}^k \lambda_i x_i^2$ and take derivative in $x$:

$$\nabla h(x) = 2(x - y) - D\Phi(x)\lambda, \tag{14}$$

where $D\Phi(x)$ is the $d \times k$ matrix given by

$$D\Phi(x) = \begin{bmatrix} 2x_1 & 0 & \ldots & 0 \\ 0 & 2x_2 & 0 & \vdots \\ \vdots & 0 & \ddots & 0 \\ 0 & \ldots & 0 & 2x_k \end{bmatrix}. \tag{15}$$

The optimal $T_\lambda(y)$ is $T_\lambda(y) = y \oslash (1 - \lambda)$ where $\oslash$ means element-wise division of vectors. If $k < d$, then complete $\lambda$ with zero entries.

**Corollary 17.** *(Quadratic case) Suppose $\Omega$ is compact, $d^2(x, y) = \|x - y\|^2$, $Q = \frac{1}{n}\sum_{i=1}^n \delta_{Z_i}$ and $\Phi(x) = (x_1^2, \ldots, x_k^2)$. Then $P^*$ is an optimal solution to problem equation P if, and only if, it is defined as $P^* = \frac{1}{n}\sum_{i=1}^n \delta_{Z_i \oslash (1-\lambda)}$ where $t = \frac{1}{n}\sum_{i=1}^n \Phi(Z_i \oslash (1 - \lambda))$.*

## E.3  LINEAR AND QUADRATIC

Take $j_0 \in \{1, \ldots, d\}$ and define $\Phi(x) = (x_{j_0}, x_{j_0}^2)$. In this case, $\Phi$ is strongly convex. If $\lambda_i < 0$ for all $i = 1, 2$ then $h(x) = \|x - y\|^2 - \sum_{i=1}^k \lambda_i \Phi_i(x)$ is strongly convex and we have existence and uniqueness of

$$\inf_x \left\{ \|x - y\|^2 - \sum_{i=1}^k \lambda_i \Phi_i(x) \right\}$$

immediately guaranteed by finding a critical point. Let $h(x) = \|x - y\|^2 - (\lambda_1 x_{j_0} + \lambda_2 x_{j_0}^2)$ and take derivative in $x$:

$$\nabla h(x) = 2(x - y) - (\lambda_1 e_{j_0} + \lambda_2 x_{j_0} e_{j_0}). \tag{16}$$

The optimal point is $T_\lambda(y) = (T_\lambda(y)_1, \ldots, T_\lambda(y)_d)$, where

$$T_\lambda(y)_i = \begin{cases} \frac{\lambda_1/2 + y_{j_0}}{1 - \lambda_2}, & \text{if } i = j_0, \\ y_i, & \text{otherwise.} \end{cases}$$

### E.4 CROSS PRODUCT

Choose $j_0, j_1 \in \{1, \ldots, d\}$ and define $\Phi(x) = x_{j_0} x_{j_1}$. For $x^* \in \mathbb{R}^d$ to be a minimizer of $h(x) = \|x - y\|^2 - \lambda x_{j_0} x_{j_1}$, it has to be a critical point. Computing the gradient

$$\frac{\partial h(x)}{\partial x_i} = 2(x_i - y_i), \text{ for } i \neq j_0, j_1 \tag{17}$$

$$\frac{\partial h(x)}{\partial x_{j_0}} = 2(x_{j_0} - y_{j_0}) - \lambda x_{j_1} \tag{18}$$

$$\frac{\partial h(x)}{\partial x_{j_1}} = 2(x_{j_1} - y_{j_1}) - \lambda x_{j_0}. \tag{19}$$

gives the following system of equations:

$$\begin{cases} x_i = y_i, \text{ for } i \neq j_0, j_1 \\ 2x_{j_0} - \lambda x_{j_1} = 2y_{j_0}, \\ 2x_{j_1} - \lambda x_{j_0} = 2y_{j_1}. \end{cases}$$

This system has a solution if and only if $\lambda \neq 2, -2$. Now, a sufficient condition for $x^*$ to be a minimizer is the positive definiteness of the hessian matrix in $x^*$. When $j_0 = 1$ and $j_1 = 2$, it is given by

$$D^2 h(x^*) = \begin{bmatrix} 2 & -\lambda & \ldots & 0 \\ -\lambda & 2 & 0 & \vdots \\ \vdots & 0 & \ddots & 0 \\ 0 & \ldots & 0 & 2 \end{bmatrix}. \tag{20}$$

In any case, the eigenvalues of $D^2 h(x^*)$ are $2 + \lambda, 2 - \lambda$ and $2$. Consequently, we need $|\lambda| < 2$.

The optimal point is $T_\lambda(y) = (T_\lambda(y)_1, \ldots, T_\lambda(y)_d)$, where

$$T_\lambda(y)_i = \begin{cases} y_i, \text{ for } i \neq j_0, j_1, \\ y_{j_0} + \frac{\lambda}{2} \frac{2y_{j_1} + \lambda y_{j_0}}{2 - \lambda^2/2}, \text{ if } i = j_0, \\ \frac{2y_{j_1} + \lambda y_{j_0}}{2 - \lambda^2/2}, \text{ if } i = j_1. \end{cases}$$

### E.5 LINEAR AND CROSS PRODUCT

Take $j_0, j_1 \in \{1, \ldots, d\}$ and define $\Phi(x) = (x_{j_0}, x_{j_1}, x_{j_0} \cdot x_{j_1})$. For $x^* \in \mathbb{R}^d$ to be a minimizer of $h(x) = \|x - y\|^2 - \lambda_1 x_{j_0} - \lambda_2 x_{j_1} - \lambda_3 x_{j_0} x_{j_1}$, it has to be a critical point. Computing the gradient

$$\frac{\partial h(x)}{\partial x_i} = 2(x_i - y_i), \text{ for } i \neq j_0, j_1$$

$$\frac{\partial h(x)}{\partial x_{j_0}} = 2(x_{j_0} - y_{j_0}) - \lambda_1 - \lambda_3 x_{j_1}$$

$$\frac{\partial h(x)}{\partial x_{j_1}} = 2(x_{j_1} - y_{j_1}) - \lambda_2 - \lambda_3 x_{j_0}.$$

gives the following system of equations:

$$\begin{cases} x_i = y_i, \text{ for } i \neq j_0, j_1 \\ 2x_{j_0} - \lambda_3 x_{j_1} = 2y_{j_0} + \lambda_1, \\ 2x_{j_1} - \lambda_3 x_{j_0} = 2y_{j_1} + \lambda_2. \end{cases} \tag{21}$$

This system has a solution if and only if $\lambda_3 \neq 2, -2$. Now, a sufficient condition for $x^*$ to be a minimizer is the positive definiteness of the hessian matrix in $x^*$. When $j_0 = 1$ and $j_1 = 2$, it is

given by

$$D^2 h(x^*) = \begin{bmatrix} 2 & -\lambda_3 & \dots & 0 \\ -\lambda_3 & 2 & 0 & \vdots \\ \vdots & 0 & \ddots & 0 \\ 0 & \dots & 0 & 2 \end{bmatrix}. \tag{22}$$

In any case, the eigenvalues of $D^2 h(x^*)$ are $2 + \lambda_3, 2 - \lambda_3$ and $2$. Consequently, we need $|\lambda_3| < 2$.

The optimal point is $T_\lambda(y) = (T_\lambda(y)_1, \dots, T_\lambda(y)_d)$, where $T_\lambda(y)_i$ is defined by

$$\begin{cases} y_i, i \neq j_0, j_1, \\ y_{j_0} + \frac{\lambda_1}{2} + \frac{\lambda_3}{2} \left( \left( y_{j_1} + \frac{\lambda_2}{2} + \frac{\lambda_3}{2}(y_{j_0} + \frac{\lambda_1}{2}) \right) \left( 1 - \frac{\lambda_3^2}{4} \right)^{-1} \right), i = j_0, \\ \left( y_{j_1} + \frac{\lambda_2}{2} + \frac{\lambda_3}{2}(y_{j_0} + \frac{\lambda_1}{2}) \right) \left( 1 - \frac{\lambda_3^2}{4} \right)^{-1}, i = j_1. \end{cases}$$

