# OpenReview forum: "Evaluating black-box vulnerabilities with Wasserstein-constrained data perturbations"
_ICLR.cc/2026/Workshop/AFAA — AFAA 2026 Poster_

### Official Review · Reviewer_QurG · 2026-02-16
**Wasserstein-projection stress tests for black-box model vulnerability and fairness “fairwashing”**

**Rating:** 4
**Confidence:** 4

**Summary:**

This paper proposes an explainability framework that actively perturbs a test distribution via Wasserstein projections under controlled constraints (e.g., shifting the mean of selected features) and then measures how black-box models’ outputs change as a function of the perturbation magnitude τ. The workflow is consistent across experiments: train once, generate projected versions of the test set over a grid of τ, then analyze response curves using proposed criteria (with α = 0.05 and 21 values of τ∈[−1,1]). The empirical section illustrates the approach on (i) Adult Income classification (feature shifts like Age/Education-Num and a “fairwashing” case where disparate impact can appear improved under distribution shift), (ii) Fashion-MNIST via brightness shifts with a stability analysis, and (iii) Boston Housing regression, reporting how mean/variance of predictions evolve under targeted feature shifts.

**Strengths:**

1. Clear, reusable experimental protocol for controlled distribution shift analysis (train → project test set by τ → plot criteria), which makes the paper easy to reproduce conceptually and compare across models/datasets.

2. The “fairwashing” example is a useful cautionary result: fairness metrics (e.g., disparate impact) can improve simply due to a mean-calibration shift that ignores the sensitive attribute, highlighting a real evaluation pitfall.

3. The appendix-style extensions (regression mean/variance criteria; prediction stability under shift; baseline comparisons to SHAP and related projection methods) add breadth and help position the method relative to common explainability tools.

**Weaknesses:**

1. The evaluation is illustrative but still somewhat limited in scope: more systematic quantitative benchmarking (e.g., across more datasets, more feature types, stronger ablations on constraint choices, and clearer guidance on selecting τ/interpreting effect sizes) would strengthen the “general” claims.

2. Some results rely heavily on qualitative curve inspection; the paper would benefit from sharper decision criteria (what constitutes “vulnerability,” “instability,” or “meaningful” fairness distortion) and sensitivity to modeling choices (e.g., categorical handling, scaling, and multi-feature constraints beyond the shown examples).

3. Practical deployment questions remain: computational cost and failure modes of solving/approximating the projection (especially for complex constraints or neural settings) could be discussed more explicitly as limitations.

---

### Official Review · Reviewer_UPif · 2026-02-19
**A creative model-agnostic approach for explainability: theoretically founded and experimentally validated, few questions for my understanding and a need to correct typos.**

**Rating:** 5
**Confidence:** 3

**Summary:**

This paper proposes a model-agnostic approach to evaluate the response of ML algorithms under constrained perturbations, in the case of Tabular and image data. They perform stress-tests of models by generating perturbations within a Wasserstein ball around the original data distribution and assessing the performance evolution. These perturbations are solutions of worst case/most influential distribution optimisation problems within the Wasserstein ball. This allows to controllably explore alternative scenarios with potentially different outputs, yielding an explainable framework for counterfactual reasoning. Furthermore, the entire approach is theoretically grounded  via explicit characterization of optimal projected distribution as push-forward measure, consistency result showing empirical projections convergence to population projections, and dual formulation connecting the projection to distributionally robust optimization. Here, what is projected is the original data distribution unto  all distributions satisfying the perturbation constraints. Finally, the approach is validated via multiple experiments on Adult Income dataset, Fashion-MNIST and "fairwashing" analysis.

**Strengths:**

1-) The paper is clear and easy to follow.
2-) The framework is mathematically well-founded.
3-) They leverage DRO and OT to creatively design a computationally tractable explainability framework. The quantile-based parametrization as a metric is also a very good idea from my point of view.
4-) Connections to previous/related works are discussed.
5-) The fairwashing concept in their fairness experiment is novel to the best of my knowledge.
6-) They were able to demonstrate practical usage of their models in the experiments section.
7-) The developed algorithm is model-agnostic, and could therefore be reused by other models in other settings.

**Weaknesses:**

1-) There are quite some typos in the present manuscript.
2-) Few questions for my understanding: Are t and t_0 used interchangeably in Eq. 5?  Also, was compactness assumed but not stated in some proofs?
3-) The choice of \alpha and its influence are not clearly discussed.
4-) Fairwashing could be used as a diagnostic tool, not as a solution as I believe that it may potentially mask the real underlying issue.
5-) The applicability to medical settings as claimed in the paper needs to be meticulously discussed. Some diagnostic features are correlated so that, varying only one may break the existing relationship and produce an improper evaluation dataset. The paper demonstrates the possibility of varying multiple features simultaneously, which is clearly a plus. Also, what do the author think about framing constraints in terms of sub-group population or conditional distributions in the case of medical applications? This could be an interesting discussion point during the workshop.

---

### Official Review · Reviewer_2U3n · 2026-02-21
**Interesting Framework for Sensitivity Analysis Under Distribution Shifts Using Wasserstein Projections. Presentation could be improved, with necessary comparisons**

**Rating:** 4
**Confidence:** 3

**Summary:**

This paper addresses the problem of studying how a model’s predictions vary under distribution shifts in the input data. The authors propose a data generation process based on a Wasserstein projection of the dataset under constraints to produce realistic, controlled distributional variations. The paper first presents the theoretical framework and derivations, followed by a parameterization of perturbations that enables their practical application. Examples of linear constraints are provided, such as brightness adjustments for images.

The experimental evaluation considers both binary classification on the Adult dataset (tabular data) and multiclass classification on FashionMNIST (image data). In the binary case, the goal is to evaluate the sensitivity of the proportion of positive predictions across different algorithms (Decision Tree, Gradient Boosting, LDA, Naive Bayes). In the multiclass case, the authors analyze the sensitivity of class prediction proportions produced by a CNN model under controlled distribution shifts.

**Strengths:**

- The sensitivity analysis of model predictions under constrained and controlled distribution shifts is an important and relevant problem.
- The use of Wasserstein distance to induce distribution changes is interesting and practical. Realistic variations of the data distribution (e.g., brightness changes in images) can be naturally defined within this framework.
- The results provide interesting and plausible insights into how model predictions vary under distribution changes.

**Weaknesses:**

1. **Clarity.**
   - Line 108: The term feasible set is used, but it is not clear what specific set it refers to in this context.

2. **Notation introduced too late.**
   Several symbols appear before they are formally defined:
   - Line 108: $k$ is defined later in the notation section.
   - Lines 232–233: $\tau$ is mentioned without prior introduction and only defined in the following section.
Proper ordering of definitions and references would improve readability.

3. **Connection between theory and experiments.**
   The theoretical analysis is sometimes difficult to follow in terms of motivation and the explanation of the theorems and derivations. In addition, the connection between the theoretical framework and the experimental sections (5.1 and 5.2) could be made more explicit. In particular, the parameter $\tau$ does not appear in the theoretical development, making it unclear how it relates to the proposed framework.

4. **Figure issues.**
   - The range of the y-axis should be consistent across figures showing proportions of predictions (Fig. 1, Fig. 2a, Fig. 4). This would facilitate visual comparison across models, attributes, and classes.
   - Figure 2a appears pixelated and is difficult to read even after zooming.
   - Figures 2b and 2c are missing captions.

5. **Missing comparisons with alternative methods.**
   Comparisons with other global explanation or sensitivity analysis methods mentioned in the related work, or with Monte Carlo sampling approaches, would be necessary to better understand the added value of the Wasserstein projection approach beyond the realistic variation.

---

### Meta-Review · Area_Chair_tWfg · 2026-02-24

**Recommendation:** Main Papers Track
**Confidence:** 4

**Metareview:**

Three reviewers have evaluated the paper with overall positive votes (summary assessment: accept/accept/strong accept).

There is a clear consensus that the paper makes valuable contributions of practical interest based on sound methodology. It is highlighted that the paper leverages compelling ideas and is well positioned in the related literature.

There are a few clarification questions and suggestions that can be addressed in preparation of the publication of the manuscript -- please note, e.g., the mentioned typos, questions regard notation, connections between sections, improvements of figures, and discussion of limitations.

Based on the reviews, I propose the paper to be accepted for the workshop.

---

### Decision · Program_Chairs · 2026-03-02

Accept (Poster)